# Inhibition of acetic acid-induced colitis in rats by new *Pediococcus acidilactici* strains, vitamin producers recovered from human gut microbiota

**Nahla M. Mansour** [1]*, **Wagiha S. Elkalla**[2], **Yasser M. Ragab**[3], **Mohamed A. Ramadan**[3]

**1** Gut Microbiology & Immunology Group, Chemistry of Natural and Microbial Products Dept., Pharmaceutical Industries Div., National Research Centre, Cairo, Egypt, **2** Microbiology and Immunology Department, Faculty of Pharmacy, Badr University in Cairo, Cairo, Egypt, **3** Microbiology and Immunology Department, Faculty of Pharmacy, Cairo University, Cairo, Egypt

* nahla_mansour@hotmail.com

**Data Availability Statement:** All relevant data are within the paper, its Supporting Information files

## Abstract

Our aim was to isolate, identify and characterize probiotic bacteria as vitamin producers in particular B2 and B9. 150 human fecal samples were collected and used for isolation of vitamin producers—probiotics. 49 isolates were chosen for screening their genome by PCR for the presence of riboflavin and folic acid genes. As a result, three isolates were selected and their production of the B2 and B9 were confirmed by HPLC. The three isolates were identified on species level by sequencing their 16S rRNA gene which showed 100% identical to strains of *Pediococcus acidilactici*. Thus, they were named as *P. acidilactici* WNYM01, *P. acidilactici* WNYM02, *P. acidilactici* WNYM03 and submitted to the Genbank database with accession numbers. They met the probiotic criteria by expressing 90–95% survival rate at pH (2.0–9.0) and bile salt up to 2% for 3 h in addition to their antimicrobial activity against gram positive and negative microorganisms. They also showed no hemolytic activity and common pattern for antibiotic susceptibility. Our three strains were tested individually or in mixture *in vivo* on rat colitis model compared to ulcerative group. The strains were administrated orally to rats in daily dose containing CFU $10^9$ for 14 days then followed by induction of colitis using acetic acid then the oral administration was continued for more four days. The histology results, the anti-inflammatory and anti-oxidative stress biomarkers showed the protective role of the strains compared to the ulcerative group. As a conclusion, we introduce novel three probiotic candidates for pharmaceutical preparations and health applications.

## Introduction

Human beings are associated with a huge number of massive varieties of microorganisms termed "microbiota" that live on and in our bodies. Its microbial communities are host specific thus the microbiota habitat the intestinal mucosal surface or even in the gut lumen are known

and GenBank under the accession numbers MW856830, MW856831 and MW865736.

**Funding:** This research work was done at the National Research Centre (Egypt).

**Competing interests:** The authors have declared that no competing interests exist.

as 'Gut Microbiota' [1–3]. The gut microbiota provides metabolites and molecules which act as fundamental elements in human health and disease.

The vitamins are essential for sustaining the proper functioning of human beings and other organisms. One of them is vitamin B complex which is a family of eight diverse vitamins that support human body for accomplishment energy from food needed for all of the basic growth and life-sustaining processes. Vitamin B deficiency could lead to many complications such as fatigue, anemia, stomach pain, numbness, depression, in addition to respiratory infection [4, 5].

B vitamins could not be synthesized by the mammals and only synthesized by certain plants and microorganisms, thus these vitamins would be supplied through the diet. The variations between the individuals in their diet, daily activity and health status leads to B vitamins deficiencies or over excess. Although, the gut microbiota is now recognized as a potential source of B vitamins however not all gut microorganisms are capable to synthesize the B vitamins. In this work we will focus on two vitamins from the B vitamins family: B2 and B9.

B2 vitamin is known as riboflavin in addition to lactoflavin as it was first isolated from milk [6, 7]. It is a water-soluble vitamin and cannot be stored in the human body thus it needs to be daily intake. B2 vitamin considers the principle growth promoting factor in vitamin B complex; it has anti-inflammatory plus antioxidant effects and defends the oxidant-mediated inflammatory damage in lungs [8] it also stimulates phagocytosis and proliferation of macrophages and neutrophils [9]. Many studies illustrate the mechanism of the anti-inflammatory effects of riboflavin [10, 11] which decrease the production of pro-inflammatory cytokines, tumor necrosis factor-$\alpha$ [TNF-$\alpha$], and interleukin-6 [IL-6]. In addition, the vitamin has a synergistic effect on the anti-inflammatory action of dexamethasone. Furthermore, it is linked to the generation of reactive oxygen species (ROS) which consider signaling key for the immune cells and inflammation via NADPH oxidase 2 [12].

Vitamin B9 is known as folate and play as a cofactor for, amino acids and DNA synthesis in addition to other metabolic reactions also it is essential for red blood cells and supports the maintenance of immunologic homeostasis. WHO/FAO [13] recommends a daily vitamin B9 intake of 400 μg for adults as its deficiency cause megaloblastic anemia [14]; the reduction of the Treg cells which control the immune responses [15]; increased susceptibility to intestinal inflammation [16] and some psychological illnesses [17]. Thus, the appropriate intake of folate is crucial for normal development, growth and health. The availability of dietary vitamin B2 and B9 in addition to the diverse microbiota to metabolize them may have impact on activation the immune system specifically the mucosal-associated invariant T (MAIT) cells in the intestine [18].

Changes in gut microbiota composition have associated with many diseases including immunity disorder, vitamins deficiency, inflammation and cancer. Strategies to improve and modulate the gut microbiota have great attention in recent years from these using probiotic microorganisms which proved several effects in health improvement [19–21]. Probiotics are defined according to [22] as live microorganisms that provide a health advantage on the host. The action of probiotic is not only species specific but also a strain specific. Here we present the isolation and characterization of three vitamin-producing strains from human gut microbiota as potential probiotics candidates capable to protect rats from induced inflammation.

## Materials and methods

### Microorganisms and growth conditions

LAB isolates were grown on de Man, Rogosa & Sharpe (MRS) agar (Laboratories Conda S.A., Madrid, Spain) at 37˚C. The indicator strains were obtained from the American Type Culture

Collection (ATCC) and cultivated as follows: *Staphylococcus aureus* (ATCC-6538) on brain heart infusion (BHI) agar (Oxoid, Basingstoke, UK); *Escherichia coli* (ATCC- 8739), *Bacillus subtilis* (ATCC-6633) and *Pseudomonas aeruginosa* (ATCC-9027) on Luria Bertani (LB) agar (Oxoid, Basingstoke, UK); *Candida albicans* (ATCC 10231) on yeast mold (YM) agar (Difco, Franklin Lakes, NJ, USA).

### Isolation of lactic acid bacteria (LAB) from human fecal samples

Fecal samples were collected from different healthy people aged up to 60 years old. The study protocol was approved by the Research Ethics Committee of Faculty of Pharmacy, Badr University in Cairo (Egypt) no. PM-101-AH and written informed consent was obtained. The samples were suspended in phosphate buffer and stored at—40˚C. Serial dilutions of each fecal sample were incubated anaerobically using the anaerobic jar and the anaerobic AnaeroGen (Oxoid Basingstoke, UK) at 37˚C on (MRS) agar (Laboratories Conda S.A., Madrid, Spain) plates containing 0.05% w/v L. cysteine (Loba Chemie, Mumbai, India). After 48 hours of incubation single colonies were picked from each plate and examined by gram staining and catalase test before storage in 40% glycerol at -40˚C [23].

### Genomic DNA extraction and PCR

Genomic DNA was extracted using the AxyPrep bacterial genomic DNA miniprep kit (Axygen Biosciences, Union City, CA, USA) from freshly overnight cultures following the manufacturer's instructions. PCR was carried out on a thermal cycler system T100 (Bio Rad, Hercules, CA, USA) using PCR Master Mix (Fermentas Life Sciences, Vilnius, Lithuania), The PCR amplifications were examined by electrophoresis purification was done using a QIA quick PCR purification kit (Qiagen, Hilden, Germany). The primers used are as listed in (Table 1) and obtained from Bioserve Company (Cairo, Egypt).

### Screening the isolates for the genes encoding the biosynthesis of folate and riboflavin

Genomic DNA isolated from the bacterial isolates were screened by PCR to detect the genes coding for the enzymes involved in the folate and riboflavin biosynthesize pathway according

**Table 1. Primers used during this study.**

| Primer name | Primer sequence 5′–3′ | Target gene | References |
|---|---|---|---|
| **Folk- F** | CCATTTCCAGGTGGGGAATC | *fol*K | [18] |
| **FolK- R** | GGGGTGGTCCAAGCAAACTT | | |
| **Folp- F** | CCASGRCSGCTTGCATGAC | *fol*P | |
| **Folp- R** | TKACGCCGGACTCCTTTTWY | | |
| **Rib A- F** | TTTACGGGCGATGTTTTAGG | *rib*A | |
| **Rib A- R** | CGACCCTCTTGCCGTAAATA | | |
| **RibB- F** | AGTAAACGGAACGGGCAAGC | *rib*B | |
| **RibB- R** | GTTGACCAGGGCACCAACTG | | |
| **Rib G- F** | TGGKAAGACGCCKCCKTGT | *rib*G | |
| **Rib G- R** | TTCACCAAYCARAATYGCTTGA | | |
| **Rib H- F** | AGGGCGAAACCGACCACTAC | *rib*H | |
| **Rib H- R** | CGATTGGGCAGTCATCGAAC | | |
| **FD1** | CCGAATTCGTCGACAACAGAGTTTGATCCTGGCTCAG | 16S rRNA | [20] |
| **RD1** | CCCGGGATCCAAGCTTAAGGAGGTGATCCAGCC | | |

to Turpin et al. [24] and the primers targeted these genes are listed in Table 1. For folate, the isolates were screened for the signature two genes; *fol*P gene encoding dihydropteroate synthase and *fol*K gene encoding 2-amino-4-hydroxy-6-hydroxymethyldihydropteridine diphosphokinase [25]. For riboflavin, isolates were screened for the four genes involved in its production *rib*A encoding 3,4-dihydroxy-2-butanone-4-phosphatesynthase, *rib*G encoding Di amino hydroxyl phosphor ribosyl amino-pyrimidine deaminase, *rib*B encoding riboflavin synthase subunit alpha, and *rib*H encoding 6,7-dimethyl-8-ribityllumazine (lumazine) synthase.

The amplified fragments corresponding to each gene were checked on agarose gel then sequenced.

## Molecular identification

The selected isolates were identified by amplification and sequencing of their 16S rRNA genes using universal primers (Table 1) [26].

## DNA sequencing and analysis

Sequencing was conducted using the dideoxy chain termination method [27] and sequence similarity was determined by the BLAST search tool within the National Centre of Biotechnology Information [28] and the sequences aligned with the CLUSTALW interface and phylogenetic tree was created in MEGA7.0 (http://www.megasoftware.net/.

## Nucleotide sequence accession numbers

The amplified sequences of the 16S rRNA gene have been deposited in the GenBank database under accession numbers.

## Acid tolerance assay

Cultures were grown in MRS broth at 37˚C for 24 hours, and then subcultured in 10 mL of MRS broth adjusted to pH 2.0. The initial bacterial concentration was determined by viable count determination on MRS; the samples were incubated for 3 hours at 37˚C. The cells were serially diluted 10-fold in PBS (pH 7.0). The residual viable count was determined by plate counting on MRS agar after 48 hours of incubation. The survival rate was calculated as the percentage of colonies grown on MRS agar compared to the initial bacterial concentration [29].

## Bile salt tolerance

The method of Gilliland et al. [30] was used to determine bile tolerance. The isolates were inoculated into MRS broths supplemented with different concentration of oxgall bile (0.3, 0.7 and 2% w/v) (Sigma-Aldrich, St. Louis, MO, USA) then incubated for 24 hours at 37˚C onto MRS agar plates. The viable count was done at 0, 3 and 24 hours and the percentage of the survival was calculated compared to the control which has no bile. The experiment was performed in triplicate.

## Susceptibility to antibiotics

Antibiotic susceptibilities for the selected isolated were tested using disc diffusion method against the following antibiotics, ampicillin, amoxicillin, chloramphenicol, kanamycin, tetracycline, ciprofloxacin, gentamycin, vancomycin, oxytetracycline, and erythromycin as described by Wayne [31]. The antibiotic discs were obtained from bioMerieux (Lyon, France).

Diameters of inhibition zones were measured in nm and compared to the Clinical and Laboratory Standards Institute (CLSI) as resistant, intermediate, and susceptible.

## Antimicrobial activity

Antimicrobial activity of the isolates against pathogenic bacteria; *E. coli*, *S. aureus* and *C. albicans* was determined by standard well diffusion method using Mueller Hinton Agar (MHA) plates [32]. The 100 ul inoculum ($10^8$ CFU/ml) of each bacterium was spread on MHA plates. Wells were made in each inoculated plate and filled with 30 μl of neutralized cell free supernatant (pH 6.5) of isolates obtained after 24 hours growth in MRS broth. Ciprofloxacin (30 mg/ml) was used as positive control and MRS broth was used as negative control. All MHA plates were incubated at 37°C for 24 hours and were examined for inhibition zones >1 mm. Each test was performed in triplicate.

## Blood hemolysis

Hemolysin activity was assessed on Columbia blood agar (Oxoid) containing 5% v/v human blood after 48 hours of incubation at 37°C [33].

## HPLC determination of riboflavin and folic acid by bacterial isolates

One ml from an overnight culture (24 hours) was centrifuged at 14,000 g for 5 min. and the supernatant was taken and analyzed by HPLC at the central laboratory of the National Research Centre, Egypt based on the method of Klejdus et al. 2004 [34]. HPLC analysis was carried out by diluting the sample 10 fold in Milli Q water and 10 μl was injected onto the HPLC, Separation was performed on a Shimadzu HPLC system with SPD-10A VP detector, a Shimadzu LC-10AD constant-flow pump, and a recorder (Shimadzu Corporation, Kyoto, Japan). The chromatographic column was LiChrospher® RP-18 (250 mm x 4.6 mm, 5 μm film thicknesses). The column was kept at room temperature a flow rate of 0.7 ml/min with a total run time of 12 min. Separation was carried out by gradient elution with Fluorite (A) and Acetonitrile containing water (75: 25) (B). Detection wave length for detection of riboflavin and folate was set at 254 nm. Riboflavin and folic acid (Loba Chemie PVT LTD) were used as standard, the quantification of the peak area was performed.

## *In vivo* evaluation of the vitamin producing- isolates in treatment of colitis

**Rats.** 36 Male wistar rats aged 6 weeks weight of about 200 mg were obtained from Cairo University and the study protocol was approved by the Research Ethics Committee of Faculty of Pharmacy, Badr University in Cairo (Egypt) no. PM-101-AH and written informed consent was obtained. The rats were kept at the animal house of the National Research Centre (NRC), Egypt and rats were handled with care. The room was at 18–20°C and 20% humidity under a 12 hours light/dark cycle and the rats were fed with commercial riboflavin-free diet and water. At the end of the experiment, animals were sacrificed by decapitation under anesthesia by well-trained person using thiopental (50 mg/kg) as demonstrated previously [35] according to AVMA Guidelines for the Euthanasia of Animals: 2020 Edition.

**Induction of colitis in rats by acetic acid.** Colitis was induced in rats by a rectal enema with 1 mL of 3.5% (v/v) acetic acid diluted in saline. Acetic acid solution was administered through a polyethylene catheter inserted into the rectum by 4.5 cm from the anus [36]. Rats without induction of colitis obtained rectal enema with an aqueous saline solution administered at the same manner as a solution of acetic acid in animals with induction of colitis.

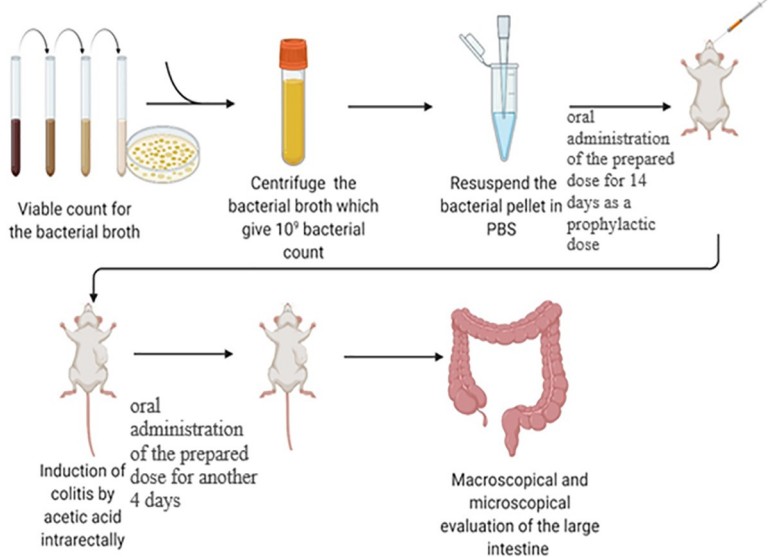

**Fig 1. Experimental design.** Rats were divided to 6 groups (n = 6 per group), the groups received the probiotic isolates or PBS daily for 14 days then on day 15 of the experiment, the groups were induced for colitis by acetic acid followed by continues the administration of the probiotics to treatment groups and PBS to the ulcerative group for 4 days before be sacrificed. The control group received PBS only without any colitis induction. The treatment groups received the isolates orally before and after the colitis induction as follows; (A), WNYM01; (B), WNYM02; (C), WNYM03; (M), Mixture of WNYM (01, 02 and 03).

**Administration of probiotic isolates and induction of colitis.** The design of the experiment is presented in Fig 1 where the oral method was used for administration of the selected strains to the experimental rats where the dose contained $10^9$ CFU in volume 200 μL of PBS. The rats were divided into six groups. Each group contained 6 rats: control group and ulcerative group received 200 μL of PBS, the treated four groups received the selected strain as follows; A: WNYM01; B: WNYM02; C: WNYM03; M: mixture of the three stains. Administration was done in daily dose for 14 days then followed by the colitis induction in day 15 for all rat groups except the control group. Then each treated group continued receiving its daily dose from selected strain and ulcerative group continued receiving PBS for 4 days and the experiment ended by sacrificing and samples were collected as previously described by Cortes-Perez et al. [37].

**Macroscopic assessment of colonic damage.** Macroscopically visible injuries such as thickening, shortening, hyperemia, and necrosis were blindly scored from 0–100% based on increasing order of severity as described by Ballester et al. [38] and Motavallian-Naeini et al. [39]. The mean scores were presented as the Disease Activity Index (DAI) of the colon.

**Microscopic assessment of colonic damage.** Samples of the distal colons were fixed immediately in 10% formaldehyde, embedded in liquid paraffin, cut into transverse sections of 5 $\mu$m thick using a Leica RM 2125 Microtome (Leica Biosystems, Wetzlar, Germany), and then mounted on glass slides and stained with haematoxylin and eosin (H&E). Microscopic changes such as necrosis, fibrosis, hyperemia, epithelial damage, ulceration, infiltration, and submucosal abscesses were scored on a 0–4 scale where 0 denotes no detectable damage and 4 denotes most severe damage [40]. Microscopy images for each group obtained with a magnification of 50X.

**Determination of the anti-inflammatory markers (IL-10 and TNF-α).** The interlukin-10 (IL-10) and Tumor Necrosis Factor Alpha (TNF-α) were tested by ELISA in the cytoplasmic extracts from homogenate rats intestinal tissues. Protein content in tissue homogenate

was determined according to the method of Bradford et al. [41]. ELISA kits "IL-10 Cloud Clone corp" (USA) and "Rat (TNF-α)" (Abbexa, UK) were used according to the manufacturer instructions. Data are represented as pg/mg total proteins and the determinations were performed in triplicate.

**Determination of lipid peroxidation marker (TBARS) and oxidative stress biomarkers (GSH).** Thiobarbituric Acid Reactive Substances (TBARS) and Glutathione (GSH) were tested by ELISA [42, 43] in the cytoplasmic extracts from homogenate rat's intestinal tissues. Kits used were TBARS Assay kit (Cayman Chemicals, USA) and Glutathione Fluorometric Assay Kit (GSH, GSSG and Total) (BioVision, USA) according to the manufacturer's instructions. Data are represented as μg/mg total proteins and the determinations were performed in triplicate.

## Statistical analysis

Data analysis was carried out using ANOVA test p value <0.05 was considered significant. Experiments were done in triplicate unless stated otherwise and all data was presented as mean ± standard deviation.

## Results

### Isolation of LAB from human feces

Initially, 150 gram-positive, catalase- negative isolates (data not shown) were selected from the incubation of the fecal samples on MRS agar plates. These isolates were evaluated for their tolerance to high acidity at pH 2.0 and alkaline conditions at pH 9.0. As a result, 49 isolates were picked for further screening according to their tolerance to low pH value by giving 80–85% survival after incubation for 3 hours at pH 2.0 in addition to their tolerance to the high pH value by showing 95% survival at pH 9.0 (Table 2).

### Screening the isolates by PCR for the presence of riboflavin and folate genes

The genomic DNA was extracted from the isolates and screened by PCR for the presence of the genes involved in the biosynthesis of each of the two vitamins riboflavin and folate. The four riboflavin genes *rib*A, *rib*B, *rib*G, and *rib*H which involved in its production were used for the detection. The PCR results (Table 2) revealed the positive of 8 isolates out of the 49 isolates by giving amplification products in the predicted sizes as follows; 121 bp, 235 bp, 351bp, 179 bp for genes *rib*A, *rib*B, *rib*G, and *rib*H respectively. The signature genes *fol*P and *fol*K were used for detection the presence of folate biosynthesis genes within the genome of the isolates, the PCR results (Table 2) showed 12 isolates out of 49 isolates are positive to the two genes by giving the amplified products in the predicted sizes; 261bp and 214 bp for *fol*P and *fol*k genes respectively. According to the PCR results, three isolates have been selected as they showed the presence of tested genes from both riboflavin and folate (Fig 2A and 2B). Each PCR fragments from the three strains were checked by sequencing and analysis by blast which revealed their identical by 98–100% to the corresponding genes from several *Pediococcus acidilactici* strains available in the database as strains with accession no: CP067392.1, CP035151.1, CP068106.1, CP028249.1, CP028247.1, CP066066.1, CP05342.1, CP018763.1, CP050079.1, CP053421.

### Molecular identification of the selected isolates

The amplified 16S rRNA products at the expected size approximately 1.5 Kb of the three isolates were checked on 1% agarose gel (Fig 3A) then were sequenced and clustered with NCBI

**Table 2. Screening the isolates for pH tolerance and the presence of folic genes (*fol*k, *fol*P) and riboflavin genes (*rib*A, *rib*B, *rib*G, *rib*H) within their genomic DNA by PCR using the specific primers as listed in Table 1.**

| Isolate No | Isolate Code | pH tolerance | | Expected PCR product size (Base Pair) for the tested genes | | | | | |
|---|---|---|---|---|---|---|---|---|---|
| | | Survival % (pH 2.00) | Survival % (pH 9.00) | *fol*k 214 bp | *fol*P 261bp | *rib*A 121 bp | *rib*B 235 bp | *rib*G 351bp | *rib*H 179 bp |
| 1 | 1AM | 85.5 ± 1.3 | 95.3 ± 2.3 | + | + | - | + | + | + |
| 2 | 1BM | 81.5 ± 1.7 | 97.1 ± 2.6 | + | + | - | + | + | + |
| 3 | 2AM | 82.3 ± 1.2 | 95.1 ± 2.3 | + | + | - | + | + | + |
| 4 | 3BM | 85. 0 ± 1.4 | 96.0 ± 2.7 | - | - | + | + | + | + |
| 5 | 3BB | 82.5 ± 1.8 | 95.1 ± 2.3 | - | - | + | + | + | + |
| 6 | 4AM | 85.0 ± 1.2 | 96.2 ± 1.8 | + | + | - | - | - | + |
| **7** | **4BM** | **85.3 ± 1.4** | **95.3 ± 1.2** | + | + | + | + | + | + |
| 8 | 4CM | 83.5 ± 1.1 | 97.1 ± 1.9 | + | + | - | + | + | + |
| 9 | SH1 | 82.3 ±1.6 | 96.3 ± 2.5 | + | + | - | - | - | - |
| 10 | SH2 | 81.7 ± 1.9 | 95.1± 2.0 | - | - | - | - | - | - |
| **11** | **SH3** | **85.0 ± 1.3** | **95.1± 2.0** | + | + | + | + | + | + |
| 12 | SH5 | 84.5 ± 1.1 | 95.1 ± 2.3 | + | + | - | + | + | - |
| 13 | SH6 | 84.5 ± 1.7 | 95.1 ± 2.3 | - | - | - | + | - | - |
| 14 | H1A | 80.4 ± 1.2 | 95.2 ± 2.1 | - | + | - | - | - | - |
| 15 | H2B | 81.2 ± 1.2 | 95.3 ± 2.0 | - | - | - | - | - | - |
| **16** | **H3C** | **85.2 ± 1.6** | **95.0 ± 1.9** | + | + | + | + | + | + |
| 17 | H4D | 80.7 ± 2.1 | 95.6 ± 2.3 | - | - | - | - | - | - |
| 18 | H5G | 80.5 ± 1.9 | 95.1 ± 2.2 | - | - | - | - | - | - |
| 19 | H6H | 80.8 ± 2.3 | 95.0 ± 1.8 | - | - | - | + | - | - |
| 20 | M6 | 85.5 ± 2.5 | 97.1 ± 2.3 | - | - | - | + | - | + |
| 21 | F1 | 81.6 ± 1.2 | 95.3± 2.4 | - | - | - | - | - | – |
| 22 | F2 | 81.5 ± 1.6 | 95.1± 2.3 | - | - | - | - | - | - |
| 23 | F7 | 82.1 ± 1.1 | 95.3 ± 2.1 | - | - | - | - | - | - |
| 24 | F8 | 82.0 ± 1.3 | 95.1 ± 2.6 | - | - | - | - | - | - |
| 25 | 9C | 85.1 ± 1.4 | 97.1 ± 2.1 | + | + | + | + | + | + |
| 26 | 8D | 84.6 ± 1.7 | 96.3 ± 2.2 | + | + | + | + | + | + |
| 27 | FA1 | 80.0 ± 1.2 | 95.9 ±1.9 | - | - | - | - | - | - |
| 28 | FA3 | 80.3 ± 1.5 | 95.5 ±2.3 | - | - | - | - | - | - |
| 29 | FA4 | 80.6 ± 1.2 | 95.6 ± 2.4 | - | - | - | - | - | - |
| 30 | FA5 | 81.0 ± 2.1 | 95.1 ± 2.8 | - | - | - | - | - | - |
| 31 | S1 | 83.5 ± 1.6 | 95.0 ± 2.9 | - | - | - | - | - | - |
| 32 | S3 | 83.8 ± 2.3 | 95.2 ± 2.1 | - | - | - | - | + | - |
| 33 | S4 | 82.9 ± 1.9 | 95.1 ± 2.3 | - | - | - | - | - | - |
| 34 | S5 | 84.6 ± 1.6 | 95.8 ± 2.9 | - | - | - | - | - | - |
| 35 | S6 | 85.1 ± 1.4 | 95.5 ± 3.3 | - | + | - | - | - | - |
| 36 | TH 1 | 83.5 ± 1.2 | 95.3 ± 2.8 | - | - | - | - | - | - |
| 37 | TH2 | 83.7 ± 1.6 | 95.0 ± 2.4 | - | - | - | - | - | - |
| 38 | TH3 | 82.9 ± 1.9 | 95.0 ± 2.5 | Two bands | 500 | 650 | 1000 | - | - |
| 39 | TH4 | 85.5 ± 2.6 | 95.2 ± 2.8 | Two bands | Two bands | - | 500 | - | - |
| 40 | TH5 | 83.5 ± 1.7 | 95.3 ±3.0 | Two bands | Two bands | - | 500 | - | - |
| 41 | TH6 | 83.4 ± 1.2 | 95.1 ± 3.2 | Two bands | - | - | - | - | - |
| 42 | SA1 | 81.9 ± 1.7 | 96.4 ± 2.5 | - | - | - | - | - | - |
| 43 | SA2 | 82.0 ± 1.3 | 95.8 ± 2.8 | - | - | - | - | - | - |
| 44 | SA3 | 82.3 ± 2.1 | 95.0 ± 2.3 | - | + | + | + | + | + |

(*Continued*)

**Table 2.** (Continued)

| Isolate No | Isolate | pH tolerance | | Expected PCR product size (Base Pair) for the tested genes | | | | | |
|---|---|---|---|---|---|---|---|---|---|
| | Code | Survival % | Survival % | *fol*K | *fol*P | *rib*A | *rib*B | *rib*G | *rib*H |
| | | (pH 2.00) | (pH 9.00) | 214 bp | 261bp | 121 bp | 235 bp | 351bp | 179 bp |
| 45 | SA4 | 83.0 ±1.2 | 96.1 ± 1.9 | - | + | - | + | - | - |
| 46 | FM1 | 85.3±1.0 | 95.1 ± 1.3 | - | - | - | - | - | - |
| 47 | FM2 | 84.9±1.3 | 97.0 ± 2.5 | - | - | + | + | + | + |
| 48 | FM3 | 85.1±1.6 | 96.2 ±2.0 | - | + | + | - | - | - |
| 49 | FM4 | 83.3±2.3 | 95.3 ±3.1 | + | - | 1000 | + | | + |

pH tolerance data represent the means ± SDs of three independent experiments.

For gene detection by PCR: (-) no bands, (+) bands in the range of the predicted size. The three selected isolates are marked in bold.

database sequences. All the three isolates showed 100% identical to *Pediococcus acidilactici* strains available in the NCBI database and phylogenetic tree was created (Fig 3B). They were named as *P. acidilactici* WNYM01, *P. acidilactici* WNYM02 and *P. acidilactici* WNYM03 and their partial sequences were submitted to GenBank under the accession numbers MW856830, MW856831 and MW865736 respectively.

## Evaluation of the probiotic criteria of the selected vitamin producing-isolates

**Bile salt tolerance.** The three isolates incubated with different concentration of bile salt and the result listed in Table 3. The three isolates showed resistance to 0.3 and 0.7% of bile salt in the MRS media up to 24 hours with a very good survival rate range from 92–97.5%. Only strain WNYM03 showed survival 73% in the 2% bile salt for 3 hours and cannot survival for longer where it shows only 15% survival after 24 hours.

**Antibiotic susceptibility.** The three isolates were tested for susceptibility to ten antibiotics listed in Table 4 using the agar diffusion method. The three showed similar pattern except for amoxicillin and Co-trimoxazole where WNYM03 was sensitive and the other two strains were intermediate. In addition, they all showed susceptibility to chloramphenicol, erythromycin, gentamycin, kanamycin and tetracycline. They all were resistant to ampicillin, ciprofloxacin and vancomycin.

**Antimicrobial activity.** The antimicrobial activity of our strains was determined using an agar well diffusion test against the pathogenic strains. As shown in Table 5 all the three strains inhibited the growth of *S. aureus*, *E. coli*, *P. aeruginosa* and *B. subtilis*, while they have no effect on *Candida albicans*.

**Blood hemolytic activity.** Hemolytic activities of the three *P. acidilactici* isolates were determined on blood agar plates. The three strains showed γ hemolytic, which means no hemolytic activity.

## HPLC detection of the riboflavin and folic acid production by selected isolates

The detection of the production of the two vitamins was done by HPLC (Table 6) where the peaks identification in the supernatant of the tested three strains was based on the retention time of the folic acid standard at 2.242 min. and the strains gave peak height as 17.27, 20.14, and 20.94% for *P. acidilactici* WNYM01, *P. acidilactici* WNYM02 and *P. acidilactici* WNYM03

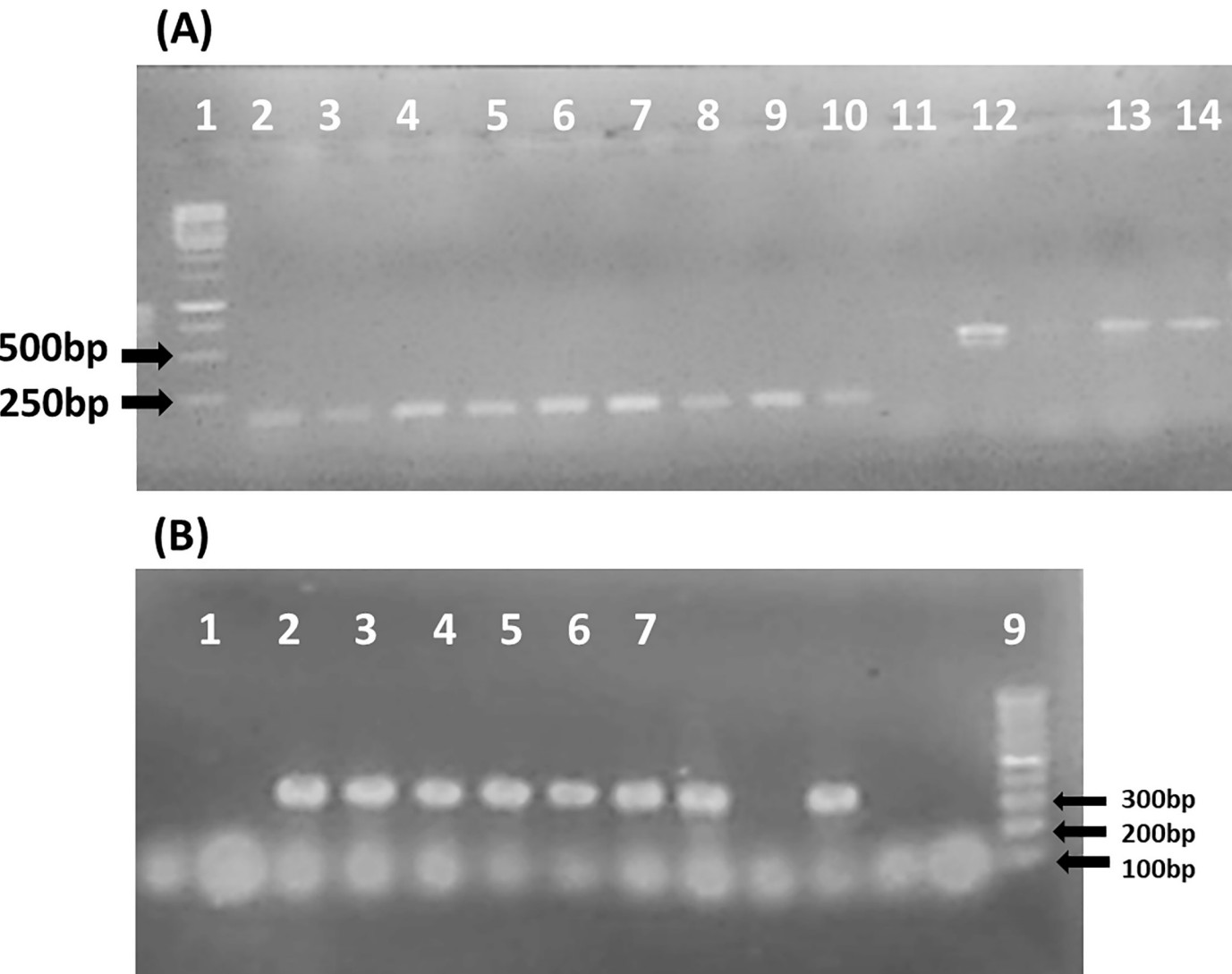

**Fig 2.** A: PCR products for detection of riboflavin and folic genes in the selected isolates using specific primers as listed in Table 1. Lane1, 1 Kb ladder; lane 2, 3, 4, PCR products for *rib*A gene from WNYM01, WNYM02, WNYM03; lane 5, 6, 7, PCR products for *rib*H gene from WNYM01, WNYM02, WNYM03; lane 8, 9, 10, PCR products for *fol*K gene from WNYM01, WNYM02, WNYM03; lane 11, negative control; lane 12, 13, 14, PCR products for *rib*G gene from WNYM01, WNYM02, WNYM03. B: Continue PCR products for detection of riboflavin and folic genes in the selected isolates using specific primers as listed in Table 1. Lane1, negative control; lane 2, 3, 4, PCR products for *rib*B gene from WNYM01, WNYM02, WNYM03; lane 5, 6, 7, PCR products for *fol*P gene from WNYM01, WNYM02, WNYM03; lane 9, 100bp ladder.

respectively. And based on the retention time of the riboflavin standard at 3.25 min where the three strains *P. acidilactici* WNYM01, *P. acidilactici* WNYM02 and *P. acidilactici* WNYM03 gave peak height as 21.19, 38.83, and 25.62% respectively.

### *In vivo* evaluation of the isolated *P. acidilactici* strains in treatment of acetic induced-colitis rats

The three isolated *P. acidilactici* strains were evaluated individually and in mixture *in vivo* for their effect on colitis in rats in four treatment groups: A (WNYM01), B (WNYM02), C (WNYM03) and M (mix of the three strains) compared to the control group and the ulcerative group. The treatment groups received our isolated *P. acidilactici* strains in daily dose before

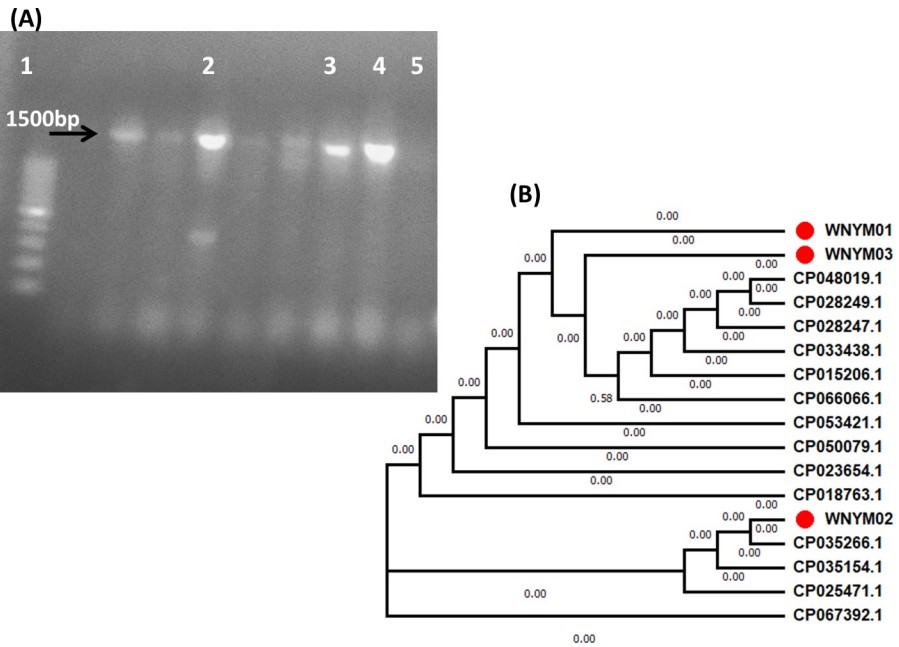

**Fig 3.** A: PCR amplification of the 16S rRNA gene from the selected isolates using universal primer set as listed in Table 1. Lane1, 100bp ladder; lane 2, WNYM01, lane 3, WNYM02; lane 4, WNYM03; lane 5, negative control. B: Phylogenetic tree of the isolated strains for molecular identification based on their 16S rRNA gene. Blast search at NCBI data base indicate their 16S rRNA gene are 100% identical to that belong to *Pediococcus acidilactici* strains which their accession numbers are listed on the tree. Phylogenetic tree was performed using MEGA7 software.

and after the induction of the colitis. Thus all the groups were critically evaluated in particular to the following points.

**Gross findings.** The gross findings in the colon segment of the experimental rats showed that; colon of control rats exhibited normal wall thickness, mucous membrane, muscular and serosal layer. On the other hand, the colon segment of the ulcerative group which did not receive any of the three isolates, displayed hemorrhagic inflammation and edematous swelling in their mucosae. The treatment groups A, B, C and M which received isolates WNYM01, WNYM02, WNYM03, and mix of the three respectively before and after the colitis induction showed healing in the ulcerative lesions in addition to normal colon segment with healthy mucosa (Fig 4).

**Microscopic examination.** The histopathological examination of rat colon in the all experimental groups exhibited the following findings:

**Table 3. Tolerance of isolated *Pediococcus acidilactici* strains to bile salts.**

| | Bile salt tolerance | | | | | |
|---|---|---|---|---|---|---|
| | Survival (%) | | Survival (%) | | Survival (%) | |
| | 0.3% bile | | 0.7% bile | | 2% bile | |
| Strain name | 3 h | 24 h | 3 h | 24 h | 3 h | 24 h |
| WNYM01 | 98.5 ± 1.0 | 96.5 ±2.1 | 95.5 ± 2.1 | 93.0± 1.4 | 73 ± 2.8 | 15.2 ± 2.5 |
| WNYM02 | 97.5 ± 1.2 | 96.0± 1.4 | 96.0 ± 1.4 | 94.0 ± 1.2 | ND | ND |
| WNYM03 | 95.5 ± 2.1 | 94 ± 1.4 | 93.5 ± 0.7 | 92.0 ± 0.7 | ND | ND |

Data represent the means ± SDs of three independent experiments. ND, not detected

**Table 4. Susceptibility of isolated *Pediococcus acidilactici* strains to antibiotics.**

| Strain name | Amp | Amx | Ery | Cip | Gen | Cm | Van | Tet | Kan | Cot |
|---|---|---|---|---|---|---|---|---|---|---|
| | 10 ug | 30 ug | 15 ug | 15 ug | 10 ug | 30 ug | 30 ug | 30 ug | 30 ug | 25 ug |
| WNYM01 | R | I | S | R | S | S | R | S | S | I |
| WNYM02 | R | I | S | R | S | S | R | S | S | I |
| WNYM03 | R | S | S | R | S | S | R | S | S | S |

Diameters of inhibition zones were measured and results expressed in terms of resistant (R), intermediate (I) and susceptible (S) according to cut off levels proposed by the NCCLS [61]. Data represent three independent experiments. Amp, ampicillin; Amx, Amoxicillin; Ery, erythromycin; Cip, ciprofloxacin; Gen, Gentamycin; Cm, chloramphenicol; Van, Vancomycin; Te, tetracycline; Kan, Kanamycin; Cot, Co-trimoxazol.

The colon segment in the control group (Fig 5) showed normal histological structure, including the mucosa (Lamina epithelialis, Lamina muscularis mucosa and lamina propria), the submucosa, the muscular coat (inner circular and outer longitudinal muscle fibers) and the serosa. On the other side in the ulcerative group (Fig 5), the colon segment illustrated multifocal mucosal ulcerations accompanied with severe tissue damage in the entire mucosal layers, in addition, severe inflammatory reaction with congestion in mucosal and submucosal blood vessels were also pronounced. Massive necrosis and desquamative changes were found in the superficial and deep epithelial cell associated with leukocyte infiltration mainly lymphocytes and a few neutrophils. The colon glands found focally deformed with no secretory activity.

The rat colon segment in the treatment groups (Fig 5) which received the three strains individually (A, B, C) or mixture of the three (M), exhibited normal histological features, as intact mucosal, submucosal, muscular and serosal layers. In addition, a healed ulcerative lesion with regeneration of the mucosal epithelium was observed in all the tested groups and the regenerated epithelial cells showed focal hyperplastic changes with hyperchromatic nuclei (regenerative signs). The mucosal proximity of the site of the healed ulcer greatly infiltrated by inflammatory cells mostly lymphocytes and plasma cells. The submucosa markedly replaced by aggregated and or follicular hyperplastic lymphoid cells.

## Assessment of inflammatory markers (TNF-α and IL-10)

TNF-α level was significantly increased in the ulcerative rat group by expressing 407.5 pg/mg compared to the control group and the treatment groups A, B, C, M by expressing 104–160 pg/mg (Fig 6A) on the other hand, IL10 was significantly decreased by expressing 121 pg/mg compared to the control group (314 pg/mg) and treatment groups (229.5–281.5 pg/mg (Fig 6B).

**Table 5. Antimicrobial activity of the isolated *Pediococcus acidilactici* strains.**

| | Inhibition Zone (mm) | | |
|---|---|---|---|
| | WNYM01 | WNYM02 | WNYM03 |
| *C. albicans* | ND | ND | ND |
| *S. aureus* | 23 ±0.05 | 24 ±0.02 | 23 ±0.05 |
| *Bacillus* | 22 ±0.02 | 22 ±0.01 | 22 ±0.03 |
| *E.coli* | 22 ±0.03 | 22 ±0.01 | 22 ±0.01 |
| *P. aeruginosa* | 25 ±0.02 | 20 ±0.02 | 20 ±0.01 |

Data represent the means ± SDs of three independent experiments. ND, not detected.

**Table 6. Detection of folic acid and riboflavin production by the isolated *P. acidilactici* strains was done using HPLC.** The culture supernatant from each strain was tested and1mg/ml of each folic acid and riboflavin was used as standard, the running volume was 10 μl, the retention time and the peak (area—height) listed.

| | Folic (1mg/ml) | WNYM01 | WNYM02 | WNYM03 |
|---|---|---|---|---|
| **Retention time** | 2.242 | 2.275 | 2.317 | 2.458 |
| **Peak Area** | 417914 | 8542069 | 8360937 | 9803561 |
| **Area %** | 64.378 | 26.121 | 25.519 | 28.925 |
| **Peak Height** | 44779 | 289023 | 288480 | 297490 |
| **Height %** | 76.081 | 17.274 | 20.141 | 20.939 |
| | Riboflavin (1mg/ml) | WNYM01 | WNYM02 | WNYM03 |
| **Retention time** | 3.250 | 3.183 | 3.217 | 3.192 |
| **Peak Area** | 2007067 | 4833985 | 11273995 | 11242746 |
| **Area %** | 86.364 | 14.782 | 34.410 | 33.171 |
| **Peak Height** | 95784 | 354613 | 398349 | 363989 |
| **Height %** | 81.438 | 21.195 | 27.811 | 25.620 |

## Assessment of lipid peroxidation marker (TBARS) and oxidative stress biomarkers (GSH)

Compared to the ulcerative rat group, both the control group and treatment groups verified significant decrease in TBARS contents (2.2–2.6 μg/mg) (Fig 7A) and in GSH contents (0.8–1.1μg/mg) (Fig 7B).

## Discussion

Vitamin B2 (Riboflavin) and B9 (folate) are essential vitamins for human health, vital metabolic process, activation the immune cells and protect from diseases. Specific species of gut microbiota play main role in supply the human body with the sustainable amount of the essential vitamins. Any unbalance in the microbiota composition or diversity could lead vitamins deficiency and metabolic diseases. Probiotics consider an effective strategy to restore the gut

**Fig 4. Gross photograph of rat's colon segment showing the appearance differences among the groups in wall thickness, mucous membrane, muscle coat and serosa.** The rat groups received the probiotic isolates or PBS daily for 14 days then induction for colitis by acetic acid on day 15 of the experiment followed by continues the administration for 4 days before be sacrificed. The control group received PBS only without any colitis induction. The treatment groups received the isolates orally before and after the colitis induction as follows; (A), WNYM01; (B), WNYM02; (C), WNYM03; (M), Mixture of WNYM (01, 02 and 03).

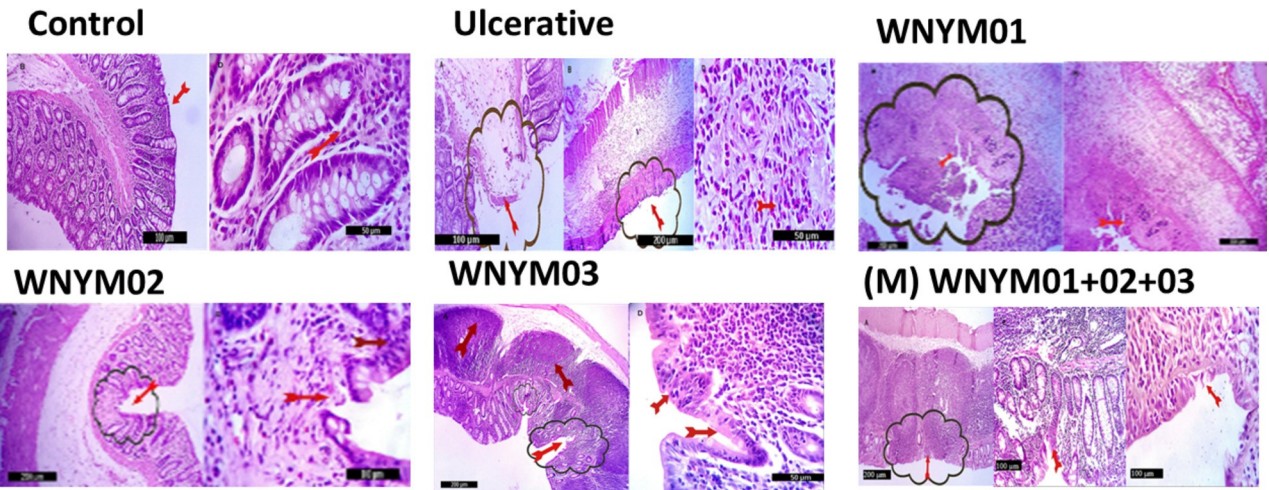

**Fig 5. Rat's colon segment showing the histo-morphological structures differences among the groups, including the mucosa (Lamina epithelialis, Lamina muscularis mucosa and lamina properia), the submucosa, the muscular coat (inner circular and outer longitudinal muscle fibers) and the serosa (arrows).** Scale bars 50 um, 100 um. The Rat groups received the probiotic isolates or PBS daily for 14 days then induction for colitis by acetic acid on day 15 of the experiment followed by continues the administration for 4 days before be sacrificed. The control group received PBS only without any colitis induction. The treatment groups received the isolates orally before and after the colitis induction as follows; (A), WNYM01; (B), WNYM02; (C), WNYM03; (M), Mixture of WNYM (01, 02 and 03).

microbiota and keep wellbeing. As the action of probiotic is strain specific, in this study we collected human fecal samples to isolate vitamin producer LAB as potential probiotics. The isolation was based on gram positive and catalase negative followed by screening the presence of biosynthesis genes; *fol*P and *fol*K for folate and *rib*A, B, G, H for riboflavin. Three isolates were chosen out of 49 isolates according to their PCR positive to the six target genes which confirmed by sequencing in addition to their tolerance to pH 2.0, the principle criteria for survival through the stomach. Their identification on the species level was based on amplification and sequencing 16S rRNA gene which was analyzed using blast research at NCBI and the results

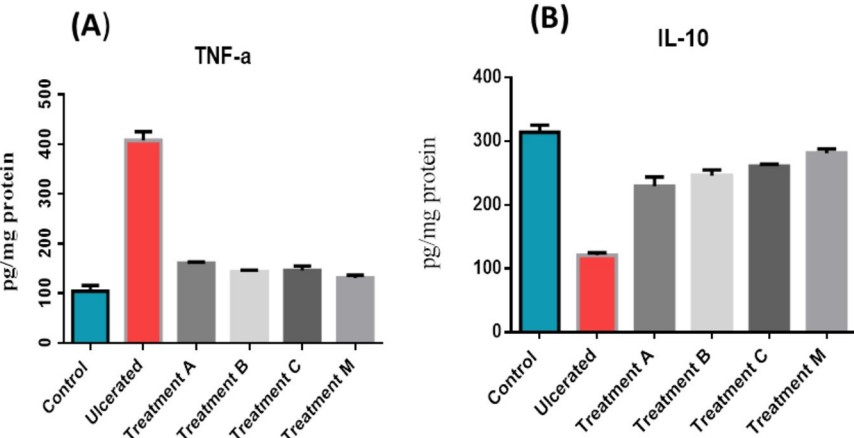

**Fig 6. Levels of inflammatory markers in the cytoplasmic extracts from homogenate rat's intestinal tissues.** (A) tumor necrosis factor-α (TNF-α) and tissue (B) interlukin-10 (IL-10). The rat experimental groups: Control, received PBS only; Ulcerative, received PBS and colitis induction; A, received WNYM01 + colitis induction; B, received WNYM02 + colitis induction; C, received WNYM03 + colitis induction. M, received Mixture of WNYM (01–03) + colitis induction. Three triplicate experiments were independently performed. Data are presented as the mean ± standard deviation.

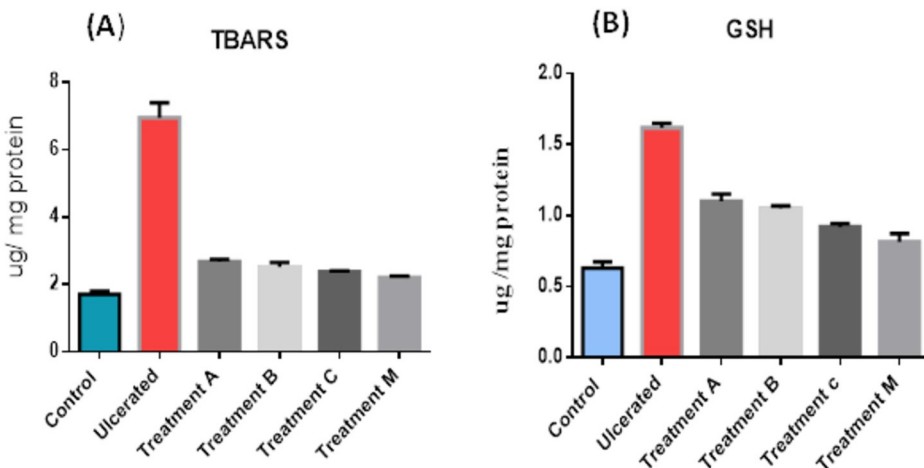

**Fig 7. Levels of lipid peroxidation marker (TBARS) and oxidative stress biomarkers (GSH) in the cytoplasmic extracts from homogenate rat's intestinal tissues.** (A), thiobarbituric acid reactive substances (TBARS) and (B), glutathione (GSH). The rat experimental groups: Control, received PBS only; Ulcerative, received PBS and colitis induction; A, received WNYM01 + colitis induction; B, received WNYM02 + colitis induction; C, received WNYM03 + colitis induction. M, received Mixture of WNYM (01–03) + colitis induction. Data are presented as mean ± S.D. (n = 3). Mean differences are significant (p < 0.05).

revealed that all of them belong to *P. acidilactici* strains available in the database by 100% identical. Our three strains named as *P. acidilactici* WNMY01, *P. acidilactici* WNMY02 and *P. acidilactici* WNMY03 met the probiotic criteria by expressing tolerance to alkaline condition (pH 9.0) 95% survival and to bile salt by giving > 90% survival to 0.7% bile salt for 24 hours and > 50% survival to 2% bile salt for 3 h these results are competing to *P. acidilactici* B14 showed 72.4% in 0.3% bile salt [44] while *P. acidilactici* SMVDUDB2 showed 80% at 0.3% - 0.5% bile salt [45]. Furthermore, our three strains showed antibacterial activity against the four tested bacteria; the gram positive (*B. subtilis* and *S. aureus*) as well as the gram negative (*E. coli* and *P. aeruginosa*). Their competitive and inhibition ability suggest them as alternative to overcome the drug resistant pathogens. This ability could be due to substances as lactic acid, hydrogen peroxide, and bacteriocins production as detected by other *P. acidilactici* strains [46, 47].

Their production of the two vitamins was detected by HPLC, thus suggested them as potential probiotics which able to colonize the gastrointestinal tract and provide the B2 and B9 vitamins to the host.

For assessment their safety, they were tested for susceptibility to ten antibiotics where they showed susceptibility or intermediate to seven antibiotics; the gram-positive spectrum antibiotic erythromycin, the broad-spectrum antibiotics tetracycline and the aminoglycosides gentamicin, amoxicillin and co-trimoxazole in addition to kanamycin and chloramphenicol these results are in agreement to previous results on LAB [48, 49]. They all were resistant only three antibiotics; vancomycin in agreement to the results obtained by Singla et al. [50] and Bhagat et al. [45] in addition to ampicillin, ciprofloxacin.

Administration of the three strains solely or as mixture to rats before and after the induction of colitis by acetic acid resulted in protection against damage and inflammation. The anti-inflammatory effect on the treated groups was proved by the assessment of inflammatory markers (TNF-α and IL-10). They showed a remarkable reduction of TNF-α by giving 160.5–131 pg/ mg compared to the ulcerative group which showed high levels 407.5±17.5 pg/ mg. The TNF-α plays a crucial roles in the pathophysiology of inflammatory bowel disease [51, 52] and has the ability to negatively affect the mucosal immune system and causing colonic

damage. These findings are in agreement with conclusion reported by Liu et al. [53]; Kanmani et al. [54]; Vincenzi et al. [55] about the action of probiotics in suppress TNF-α release. On the other hand, the three strains and their mixture showed significant increase in the IL-10, the best key anti-inflammatory cytokine in the immune response [56] as they released IL-10 as 281.5–229.5 pg/mg compared to the ulcerative group which showed low level of IL-10 as 121 ±4.0 pg/mg.

There is increase evidence that colonic inflammation, both in human patients and experimental animals, is related to increased release of reactive oxygen metabolites and nitric oxide levels. The assessment of lipid peroxidation marker (TBARS) and oxidative stress biomarkers (GSH) in rats tissues revealed the action of our three strains and their mixture in significant reduction of their value to (2.2–2.6 μg/mg) TBARS and (0.8–1.1μg/mg) GSH compared to the ulcerative group which release 6.95±0.45 μg/mg TBARS and 1.617±0.03 μg/mg GSH. This reduction in the TBARS and GSH markers could have protective effects against inflammation. Similar results reported by Watanabe et al., 2020 [57] as concluded the effective of *L. plantarum* P1-2 and *P. pentosaceus* Be1 for reducing oxidative stresses *in vivo*.

The anti-inflammatory properties of the three strains could be also due to the riboflavin and folic acid effects. Different studies reported the anti-inflammatory effect of riboflavin and folic acid. Leblanc et al., 2020 [58] and the ability of folic acid and riboflavin producer LAB to treat inflammation in animal models was reported by Levit et al., 2017 [59] and Levit et al., 2020 [60].

## Conclusions

Here we represent the isolation, characterization and in vivo evaluation of *P. acidilactici* WNYM01, WNYM02 and WNYM03 strains which containing the screened biosynthesis genes for vitamin B2 and B9. The administration of our three strains of *P. acidilactici* to experimental rat groups before and after the induction of colitis prevent tissues damage and was associated with a reduction of inflammatory and oxidative stress biomarkers, while increasing the vital cytokine for immune response. These findings are suggesting potential novel probiotic candidates for modulation the gut microbiota and pharmaceutical industries. However, complete genome sequence for these valuable strains is in progress.

## Supporting information

**S1 Table. The antibiotics susceptibility of the isolates; WNYM01, WNYM02 and WNYM03.**
(DOCX)

**S2 Table. Assessment of inflammatory markers in the cytoplasmic extracts from homogenate rat's intestinal tissues tumor necrosis factor-α (TNF-α) and intelukin-10 (IL-10).** The rat experimental groups: control, received PBS only; Ulcerative, received PBS and colitis induction; A, received WNYM01 + colitis induction; B, received WNYM02 + colitis induction; C, received WNYM03 + colitis induction. M, received Mixture of WNYM (01–03) + colitis induction. Data are presented as mean ± S.D. (n = 3). *Significance compared to Ulcerative. Mean differences are significant (p < 0.05).
(DOCX)

**S3 Table. Assessment the levels of lipid peroxidation marker thiobarbituric acid reactive substances (TBARS) and oxidative stress biomarkers glutathione (GSH) in the cytoplasmic extracts from homogenate rat's intestinal tissues.** The rat experimental groups: control, received PBS only; Ulcerative, received PBS and colitis induction; A, received WNYM01

+ colitis induction; B, received WNYM02 + colitis induction; C, received WNYM03 + colitis induction. M, received Mixture of WNYM (01–03) + colitis induction. Data are presented as mean ± S.D. (n = 3). * Significance compared to Ulcerative. Mean differences are significant (p < 0.05).
(DOCX)

## Author Contributions

**Conceptualization:** Nahla M. Mansour.

**Data curation:** Nahla M. Mansour, Wagiha S. Elkalla.

**Formal analysis:** Nahla M. Mansour, Wagiha S. Elkalla, Yasser M. Ragab, Mohamed A. Ramadan.

**Investigation:** Nahla M. Mansour, Wagiha S. Elkalla.

**Methodology:** Wagiha S. Elkalla.

**Project administration:** Nahla M. Mansour, Yasser M. Ragab, Mohamed A. Ramadan.

**Resources:** Nahla M. Mansour.

**Software:** Wagiha S. Elkalla.

**Supervision:** Nahla M. Mansour, Yasser M. Ragab, Mohamed A. Ramadan.

**Validation:** Nahla M. Mansour, Wagiha S. Elkalla, Yasser M. Ragab, Mohamed A. Ramadan.

**Visualization:** Nahla M. Mansour, Wagiha S. Elkalla.

**Writing – original draft:** Nahla M. Mansour, Wagiha S. Elkalla.

**Writing – review & editing:** Nahla M. Mansour, Yasser M. Ragab, Mohamed A. Ramadan.

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
