## [Decision Letter · Decision Letter 0]

10 Jun 2021

PONE-D-21-15668

Inhibition of Acetic acid-Induced Colitis in Rats by P. acidilactici WNYM (01-03); Vitamin Producers Recovered from  Human Gut Microbiota

PLOS ONE

Dear Dr. Mansour,

Thank you for submitting your manuscript to PLOS ONE. After careful consideration, we feel that it has merit but does not fully meet PLOS ONE’s publication criteria as it currently stands. Therefore, we invite you to submit a revised version of the manuscript that addresses the points raised during the review process.

We look forward to receiving your revised manuscript.

Kind regards,

Mahmoud Abdel Aziz Mabrok, PhD

Academic Editor

PLOS ONE

Journal Requirements:

Reviewers' comments:

Reviewer's Responses to Questions

**Comments to the Author**

1. Is the manuscript technically sound, and do the data support the conclusions?

Reviewer #1: Yes

Reviewer #2: Yes

2. Has the statistical analysis been performed appropriately and rigorously? 

Reviewer #1: No

Reviewer #2: Yes

3. Have the authors made all data underlying the findings in their manuscript fully available?

Reviewer #1: Yes

Reviewer #2: No

4. Is the manuscript presented in an intelligible fashion and written in standard English?

Reviewer #1: Yes

Reviewer #2: Yes

5. Review Comments to the Author

Reviewer #1: Inhibition of Acetic acid-Induced Colitis in Rats by P. acidilactici WNYM (01-03); Vitamin Producers Recovered from Human Gut Microbiota

The manuscript should be revised for proper used of abbreviations.

Do not start paragraphs with abbreviations or mathematical numbers.

The ethical approval number should be written in details.

Animal handling, anesthesia and euthanasia should be written in details.

How many animals per group?

Why not used one-way ANOVA for statistical analysis to compare among the studied groups.

The pathogenesis of ulcerative colitis in relation to oxidative stress and inflammation should be discussed in more details. The following reference might be helpful:

https://doi.org/10.3109/08923973.2014.998368

The biomedical applications of probiotic should be discussed in more details. The following references might be helpful.

Biomolecules 2021, 11(5), 678; https://doi.org/10.3390/biom11050678

TBARS is a lipid peroxidation marker (not oxidative stress marker). Please discuss.

For oxidative stress markers, why not chose enzymatic antioxidant markers (SOD, CAT or GSH-Px). The authors chose only reduced glutathione (GSH).

I recommend immunohistochemistry of colon tissue (COX-2, Caspase 3, INOS, BCL2, BAX)

Why the authors chose IL10 with TNF-α Why not IL-1beta, IL-6 as proinflammatory cytokines.

Reviewer #2: Major comments:

1- Lack of references in many paragraphs inside the manuscript as follows:

- 2nd paragraph in the Introduction section includes many information without references.

- Last line in page 3 (many studies ??) where are the references of these studies.

2- Methods section also there are many references lacking as follows:

- Acid Tolerance assay ??

- Blood hemolysis

- HPLC for determination of riboflavin & folate

- Determination of oxidative stress biomarkers

3- Expression of the results in the form of tables or figures are necessary & more impressive as text only is not enough as follows:

- Screening the isolation by PCR

- Photo of Gel electrophoresis Band.

- Molecular identification

- HPLC peak point & retention time

4- Gross finding: because you administered the bacterial strains for 14 days before colitis induction, So, this is not treatment regimen, it is prophylactic or preventive strategy. If you administer the strain after colitis induction start, at this case become treatment regimen.

5- Discussion section: there is no discussion for the antibacterial activity of the bacterial strains??

6- Also, the results of antioxidant activity of the bacterial strain are not discussed & no interpretation has mentioned.

Minor corrections:

a- Abbreviation in the title is not preferred (P. Pediococcus)

b- First sentence in the second paragraph in introduction section to be revised.

c- Introduction: 2nd paragraph: Vitamin B lack (un common word) to be changed in to Vitamin B deficiency; Also respiratory contagion to be changed to respiratory infection.

d- Page (4), second paragraph, second sentence is very long need to be revised as not logic to contain (and & furthermore) in the same sentence.

e- Page (10): first paragraph last line, the word after to change to "Later".

f- Result section: isolation of LAB: no need to repeat the first 5 lines as it is mentioned in the method section & the results start definitely from the line 6.

6. PLOS authors have the option to publish the peer review history of their article (what does this mean?). If published, this will include your full peer review and any attached files.

Reviewer #1: No

Reviewer #2: **Yes: **Abdelfattah Mohamed Abdelfattah Ali

---

## [Author Response · Author response to Decision Letter 0]

6 Jul 2021

Dear Editor of PLOS ONE, 

Dear Reviewers. 

Thank you for your valuable comments, which is very helpful 

the manuscript has been revised accordingly ( any changes in the text, references, tables, figures are highlighted in green)

Please find my point by point reply on the comments in blue 

Reviewer #1: 

Inhibition of Acetic acid-Induced Colitis in Rats by P. acidilactici WNYM (01-03); Vitamin Producers Recovered from Human Gut Microbiota

The manuscript should be revised for proper used of abbreviations.Do not start paragraphs with abbreviations or mathematical numbers.

The title Changed to: Inhibition of Acetic acid-Induced Colitis in Rats by New Pediococcus acidilactici Strains, Vitamin Producers Recovered from Human Gut Microbiota 

The ethical approval number should be written in details.

The ethical approval number is added 

Animal handling, anesthesia and euthanasia should be written in details.

It is added and reference also is added as follows

At the end of the experiment, animals were sacrificed by decapitation under anesthesia by well-trained person using thiopental (50 mg/kg) as demonstrated previously according to AVMA Guidelines for the Euthanasia of Animals: 2020 Edition.

Atef RM, Agha AM, Abdel-Rhaman AA, Nassar NN. The Ying and Yang of Adenosine A1 and A2A Receptors on ERK1/2 Activation in a Rat Model of Global Cerebral Ischemia Reperfusion Injury. Mol Neurobiol. 2018 Feb;55(2):1284-1298. doi: 10.1007/s12035-017-0401-1. Epub 2017 Jan 24. PMID: 28120151.

How many animals per group?

It is already indicated in the method section page 9 as below under the title 

Administration of probiotic isolates and induction of colitis

The design of the experiment is presented in Figure (1) where the oral method was used for administration of the selected strains to the experimental rats where the dose contained 109 CFU in volume 200 μL of PBS. The rats were divided into six groups. Each group contained 6 rats:

Why not used one-way ANOVA for statistical analysis to compare among the studied groups.

We changed to ANOVA 

The pathogenesis of ulcerative colitis in relation to oxidative stress and inflammation should be discussed in more details. The following reference might be helpful:

https://doi.org/10.3109/08923973.2014.998368

we extend the discussion in this part

The biomedical applications of probiotic should be discussed in more details. The following references might be helpful.

Biomolecules 2021, 11(5), 678; https://doi.org/10.3390/biom11050678

very helpful papers indeed, 

TBARS is a lipid peroxidation marker (not oxidative stress marker). Please discuss.

For oxidative stress markers, why not chose enzymatic antioxidant markers (SOD, CAT or GSH-Px). The authors chose only reduced glutathione (GSH).

I recommend immunohistochemistry of colon tissue (COX-2, Caspase 3, INOS, BCL2, BAX)

- We changed the title to Lipid peroxidation within the manuscript

- However; Lipid peroxidation is a sensitive marker due to the high likelihood of lipids to undergo oxidation, and therefore is a highly used oxidative stress marker (Lavie et al. 2004). 

- L. Lavie, A. Vishnevsky, P. Lavie Evidence for lipid peroxidation in obstructive sleep apnea Sleep, 27 (2004), pp. 123-128

- Determination of oxidative stress TBARS are produced during lipoperoxidation oxidative stress induced damage of lipids, and are thus a widely used marker of oxidative stress [4,11]

- According to the budget situation, we chose only these markers 

- For immunohistochemistry of colon tissue, it is a good recommendation we could work on it in the next paper 

- Why the authors chose IL10 with TNF-α Why not IL-1beta, IL-6 as proinflammatory cytokines.

- We concentrated in these two for their importance in addition we have limited budget thus we cannot expand more 

Reviewer 2 

Dear Editor of PLOS ONE

Please, find the review of the manuscript PONE-D-21-15668 entitled "Inhibition of Acetic acid-Induced Colitis in Rats by P. acidilactici WNYM (01-03); Vitamin Producers Recovered from Human Gut Microbiota"

Major comments:

1- Lack of references in many paragraphs inside the manuscript as follows:

- 2nd paragraph in the Introduction section includes many information without references.

- References are added

- Last line in page 3 (many studies ??) where are the references of these studies.

- References are added

2- Methods section also there are many references lacking as follows:

- Acid Tolerance assay ?? 

- References is added

- Blood hemolysis

- Reference is added

- HPLC for determination of riboflavin & folate

- Reference is added

- Determination of oxidative stress biomarkers

 References are added 

3- Expression of the results in the form of tables or figures are necessary & more impressive as text only is not enough as follows:

- Screening the isolation by PCR

- Table 2 was added which show the screening of the isolation by PCR and pH tolerance

- Photo of Gel electrophoresis Band. 

- Figure 2 A-B was added to show the gel electrophoresis bands for the three selected isolates 

- Molecular identification

- Figure 3 A was added to show the gel electrophoresis bands for the 16Sr RNA amplification fragment from the three isolates

- Figure 3B was added which show the phylogenetic tree for molecular identification 

- HPLC peak point & retention time 

- Table 3 was added to show the HPLC peak point & retention time

4- Gross finding: because you administered the bacterial strains for 14 days before colitis induction, So, this is not treatment regimen, it is prophylactic or preventive strategy. If you administer the strain after colitis induction start, at this case become treatment regimen. 

Actually the bacterial strains were administrated 14 days before colitis induction and 4 days after colitis induction. 

The method part has been changed for clarification this point. 

5- Discussion section: there is no discussion for the antibacterial activity of the bacterial strains??

The discussion was added for this part

6- Also, the results of antioxidant activity of the bacterial strain are not discussed & no interpretation has mentioned.

The discussion and interpretation was added for this part

Minor corrections:

a- Abbreviation in the title is not preferred (P. Pediococcus)

The title is Changed to: 

Inhibition of Acetic acid-Induced Colitis in Rats by New Pediococcus acidilactici Strains, Vitamin Producers Recovered from Human Gut Microbiota 

First sentence in the second paragraph in introduction section to be revised.

It is revised to: The vitamins are essential for sustaining the proper functioning of human beings and other organisms. 

b- Introduction: 2nd paragraph: Vitamin B lack (un common word) to be changed in to Vitamin B deficiency; Also respiratory contagion to be changed to respiratory infection.

It changed to: Vitamin B deficiency

It changed to: Respiratory infection

c- Page (4), second paragraph, second sentence is very long need to be revised as not logic to contain (and & furthermore) in the same sentence.

It is revised as follows:

WHO/FAO [13] recommends a daily vitamin B9 intake of 400 μg for adults as its deficiency cause megaloblastic anemia [14]; the reduction of the Treg cells which control the immune responses [15]; increased susceptibility to intestinal inflammation [16] and some psychological illnesses [17].

d- Page (10): first paragraph last line, the word after to change to "Later". 

 The entire sentence is changed to: 

Then each group continued its daily dose in the same pattern as before the colitis induction for 4 days and the experiment ended by sacrificing and samples were collected. 

e- Result section: isolation of LAB: no need to repeat the first 5 lines as it is mentioned in the method section & the results start definitely from the line 6.

Changed to: 

Initially, 150 gram-positive, catalase- negative isolates (data not shown) were selected from the incubation of the fecal samples on MRS agar plates. These isolates were evaluated for their tolerance to high acidity at pH 2.0 and alkaline conditions at pH 9.0. As a result, 49 isolates were picked for further screening according to their tolerance to low pH value by giving 80-85% survival after incubation for 3 hours at pH 2.0 in addition to their tolerance to the high pH value by showing 95 % survival at pH 9.0

---

## [Decision Letter · Decision Letter 1]

12 Jul 2021

Inhibition of Acetic acid-Induced Colitis in Rats by New Pediococcus acidilactici Strains, Vitamin Producers Recovered from Human Gut Microbiota

PONE-D-21-15668R1

Dear Dr. Mansour,

We’re pleased to inform you that your manuscript has been judged scientifically suitable for publication and will be formally accepted for publication once it meets all outstanding technical requirements.

Kind regards,

Mahmoud Abdel Aziz Mabrok, PhD

Academic Editor

PLOS ONE

Additional Editor Comments (optional):

Based on the reviewers' comment, I would like to appreciate your efforts to present the manuscript in the current version. The authors addressed all scientific comments and inquiries as required, and therefore the final decision is acceptable. However, based on my review, I would like to suggest adding a small paragraph on MDR (multi-drug resistance and its negative impact on the animal industry, which also reflects the potential use of certain probiotics as an alternative compound for disease prevention and control. Please use the following valuable manuscripts as a guide for further editing. They discussed briefly the emergence of MDR in different animals, including poultry, cattles, fish

1-PMID: 32235800

2- PMID: 32497922

3-PMID: 32472209 DOI: 10.1186/s13568-020-01037-z

4-PMID: 33061472

5- https://doi.org/10.1016/j.aquaculture.2020.736070

Reviewers' comments:

Reviewer's Responses to Questions

**Comments to the Author**

1. If the authors have adequately addressed your comments raised in a previous round of review and you feel that this manuscript is now acceptable for publication, you may indicate that here to bypass the “Comments to the Author” section, enter your conflict of interest statement in the “Confidential to Editor” section, and submit your "Accept" recommendation.

Reviewer #1: All comments have been addressed

Reviewer #2: All comments have been addressed

2. Is the manuscript technically sound, and do the data support the conclusions?

Reviewer #1: Yes

Reviewer #2: Yes

3. Has the statistical analysis been performed appropriately and rigorously? 

Reviewer #1: Yes

Reviewer #2: Yes

4. Have the authors made all data underlying the findings in their manuscript fully available?

Reviewer #1: Yes

Reviewer #2: Yes

5. Is the manuscript presented in an intelligible fashion and written in standard English?

Reviewer #1: (No Response)

Reviewer #2: No

6. Review Comments to the Author

Reviewer #1: As The authors addressed the reviewers comments, I suggest acceptance of the manuscript. The manuscript is acceptable in the current form.

Reviewer #2: No more comments as the manuscript has been revised correctly, language is carefully written clear & correct; No topographical errors present and no specific errors

7. PLOS authors have the option to publish the peer review history of their article (what does this mean?). If published, this will include your full peer review and any attached files.

Reviewer #1: No

Reviewer #2: **Yes: **Abdelfattah Mohamed Abdelfattah Ali

---

## [Editor Report · Acceptance letter]

16 Jul 2021

PONE-D-21-15668R1 

Inhibition of Acetic acid-Induced Colitis in Rats by New *Pediococcus acidilactici* Strains, Vitamin Producers Recovered from Human Gut Microbiota 

Dear Dr. Mansour:

I'm pleased to inform you that your manuscript has been deemed suitable for publication in PLOS ONE. Congratulations! Your manuscript is now with our production department. 

Kind regards, 

on behalf of

Dr. Mahmoud Abdel Aziz Mabrok 

Academic Editor

PLOS ONE